# Electrical activity controls area-specific expression of neuronal apoptosis in the mouse developing cerebral cortex

Oriane Blanquie, Jenq-Wei Yang, Werner Kilb, Salim Sharopov, Anne Sinning[†], Heiko J Luhmann[†]*

Institute of Physiology, University Medical Center of the Johannes Gutenberg University Mainz, Mainz, Germany

**Abstract** Programmed cell death widely but heterogeneously affects the developing brain, causing the loss of up to 50% of neurons in rodents. However, whether this heterogeneity originates from neuronal identity and/or network-dependent processes is unknown. Here, we report that the primary motor cortex (M1) and primary somatosensory cortex (S1), two adjacent but functionally distinct areas, display striking differences in density of apoptotic neurons during the early postnatal period. These differences in rate of apoptosis negatively correlate with region-dependent levels of activity. Disrupting this activity either pharmacologically or by electrical stimulation alters the spatial pattern of apoptosis and sensory deprivation leads to exacerbated amounts of apoptotic neurons in the corresponding functional area of the neocortex. Thus, our data demonstrate that spontaneous and periphery-driven activity patterns are important for the structural and functional maturation of the neocortex by refining the final number of cortical neurons in a region-dependent manner.

DOI: https://doi.org/10.7554/eLife.27696.001

*For correspondence: luhmann@uni-mainz.de

[†]These authors contributed equally to this work

**Competing interests:** The authors declare that no competing interests exist.

## Introduction

The assembly of neural circuits in the developing mammalian brain is characterized by an excess generation of neurons and elimination of a substantial portion of these neurons by apoptosis. A first wave of apoptosis, peaking at embryonic day (E) 14 in the rodent cortex, is believed to select the appropriate neural clones (*Blaschke et al., 1996*, *1998*). A second surge of apoptosis affects the early postnatal cortex, triggering the loss of up to 50% of newly differentiated neurons (*Nikolić et al., 2013*; *Dekkers et al., 2013*; *Southwell et al., 2012*). Developmental cell death therefore represents an important process contributing to the establishment of the final number of neurons (*Haydar et al., 1999*) and ultimately to the establishment of high order complex behavior (*Nakamura et al., 2016*; *Fang et al., 2014*).

The death of developing neurons is mediated by the evolutionary conserved mitochondrial pathway (*Shi, 2001*). In the mammalian brain, this intracellular pathway is highly regulated by proteins of the Bcl-2 family (*Southwell et al., 2012*; *Ledonne et al., 2016*; *Nakamura et al., 2016*; *Shindler et al., 1997*). Pro-survival factors suppress the cell death machinery and thereby prevent apoptosis (*Léveillé et al., 2010*; *Davies, 2003*). In the peripheral nervous system, neurotrophins represent the main pro-survival factors by activating the PI3K/Akt and Raf/MEK/ERK pathways, while inhibiting the JNK pathway (*Kristiansen and Ham, 2014*). Thus, competition for the neurotrophins released by the innervated targets ensures the matching between number of innervating neurons and size of their targets (*Levi-Montalcini, 1987*; *Barde, 1989*). However, experiments using transgenic mice have shown that this so-called 'neurotrophic theory' does not apply to the central nervous system (*Jones et al., 1994*; *Minichiello and Klein, 1996*; *Dekkers et al., 2013*; *Silos-*

*Santiago et al., 1997*). In developing cortical neurons, the intracellular Akt/PI3K pathway is of particular importance to promote neuronal survival (*Wagner-Golbs and Luhmann, 2012*; *Golbs et al., 2011*; *Murase et al., 2011*; *Datta et al., 1997*; *Bittigau et al., 2003*), but the upstream factors defining the fate of single developing neurons in vivo remain unknown. In general, an increase in spontaneous activity is commonly associated with a decreased rate of apoptosis whereas a reduction of electrical activity exacerbates the number of central neurons undergoing apoptosis (*Golbs et al., 2011*; *Murase et al., 2011*; *Ikonomidou et al., 1999*; *Lebedeva et al., 2017*). On the other hand, transplantation studies with embryonic GABAergic interneuron populations suggest that both the extent and time point of cell death in cortical inhibitory neurons are intrinsically determined (*Southwell et al., 2012*).

To address the question whether neuronal identity and/or neuronal activity determines the cell fate of developing cortical neurons, we performed in vivo and in vitro experiments in early postnatal mouse cerebral cortex using immunohistochemical and electrophysiological methods. These experiments demonstrate that during a defined time window of early postnatal development, the neocortex shows clear layer- and area-disparities in the number of apoptotic neurons. By comparative analysis of two functionally distinct cortical areas displaying particularly striking differences in the number of apoptotic neurons, the primary motor cortex (M1) and primary somatosensory cortex (S1), we demonstrate that electrical activity is essential to control the number of apoptotic versus surviving neurons in an area-specific manner. We further report that somatosensory deprivation in vivo leads to exacerbated amounts of degenerating neurons specifically in the corresponding sensory cortical area, demonstrating for the first time that electrical activity affects the final number of central neurons in order to match the size of functional neuronal networks.

## Results

### Developmental apoptosis affects cortical neurons in an age-, area- and layer-dependent manner

We first characterized the temporal profile of cell death in the developing neocortex of wild-type mice over the first two postnatal weeks. For that purpose, apoptotic cells were labeled with an antibody against activated-Caspase3 (aCasp3), an early marker of irreversible entry in the apoptosis process (*Jänicke et al., 1998*), and the density of aCasp3-positive cells in different neocortical areas was quantified. In agreement with previous reports (*Southwell et al., 2012*; *Verney et al., 2000*), the overall rate of cell death in the developing mouse neocortex increases from P2 on, peaks around P5 to P6 and subsequently declines (*Figure 1A* and *Figure 1—source data 1*; one-way ANOVA, $R^2$=0.54, p=0.002, n = 38 mice). Cell death at this developmental stage almost exclusively affects GFAP-negative, neuronal cells (*Figure 1B*). Consistent with previous work (*Southwell et al., 2012*; *Sahara et al., 2012*), the majority of aCasp3-positive neurons are GAD67-negative, glutamatergic neurons with a minor contribution of GAD67-positive, GABAergic neurons (*Figure 1C and D*).

To obtain a comprehensive spatial profile of cell death within the neocortex, we next tangentially sectioned and flattened cortical hemispheres of P5-7 wild-type mice. Functional cortical areas were identified in flat mounts by immunhistochemical staining against the serotonin (5-HT) transporter (*Leingärtner et al., 2007*) and apoptotic cells were co-labeled by aCasp3 (*Figure 2A*). Although the whole cortex is populated by apoptotic cells at this age, strong area discrepancies become evident. Most obviously, the primary motor cortex (M1) shows higher densities of aCasp3-positive cells (39.66 ± 14.93 aCasp3/mm$^2$) compared to the primary somatosensory cortex (S1) (15.51 ± 6.90 aCasp3/mm$^2$; n = 3 mice), although the difference is not significant (t-test, p=0.28). For a comprehensive, comparative analysis of cortical cell death in these two areas, we used coronal brain sections containing both regions (*Figure 2B*). The cortical hemispheres in these coronal sections were divided into six equal sectors - with medial sectors a and b corresponding to M1 and more lateral sectors c to f to S1. The relative density of aCasp3-labeled cells per sector was quantified in layers I to IV, layer V and layer VI. In most cases, aCasp3-positive neurons are located in layer II at the border of layer I (*Figure 2C*). Note that aCasp3 labeling allows the detection of apoptotic neurons at various stages of morphological and nuclear degradation.

Quantification of the number of aCasp3-labeled neurons revealed a higher rate of cell death in superficial cortical layers (*Figure 3A* and *Figure 3—source data 1*; layers I-IV: 29.31 ± 4.13 aCasp3/

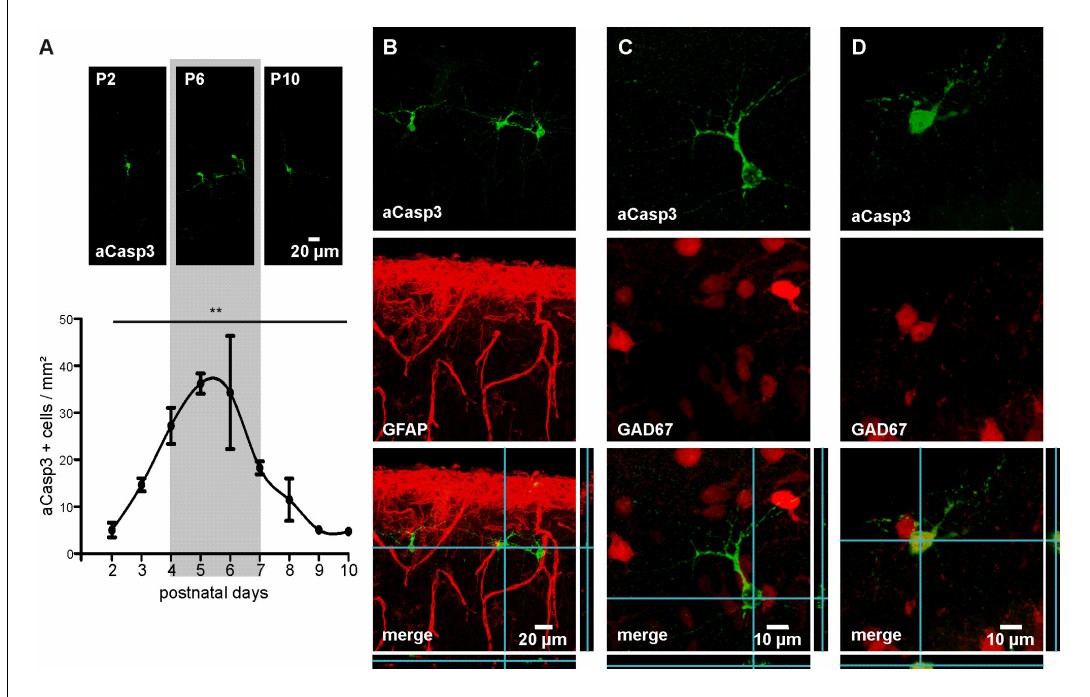

**Figure 1.** Characterization of neuronal apoptosis in the developing mouse neocortex in vivo. (A) Representative images of aCasp3-stained cells in 50 µm thick coronal slices of mouse pups and quantitative analysis of the density of apoptotic cells in the mouse primary motor and somatosensory cortices show that cell death peaks between postnatal day (P) 4 and 7 (grey shaded area) and declines until P9-10. (B) aCasp3 labeling (green) does not colocalize with the astrocytic marker GFAP (red). (C) Most of aCasp3-positive neurons (green) are GAD67-negative, putative glutamatergic neurons. (D) A subset of apoptotic neurons is expressing the GABAergic marker GAD67 (red and green).

DOI: https://doi.org/10.7554/eLife.27696.002

The following source data is available for figure 1:

**Source data 1.** Quantification of aCasp3-positive cells/mm$^2$ in M1 and S1 regions of newborn mice.
DOI: https://doi.org/10.7554/eLife.27696.003

mm$^2$) compared to deeper layers (layer V: 18.73 ± 1.61 aCasp3/mm$^2$; layer VI; 6.76 ± 0.85 aCasp3/mm$^2$; two-way ANOVA layer p=0.0002; post-hoc test sector a p<0.001, sectors b to f p>0.05; n = 11 mice). Furthermore, cell death is four times higher in M1 (*Figure 3A*; sectors a/b: 57.76 ± 10.26 aCasp3/mm$^2$) than in S1 (sectors c-f: 14.86 ± 1.52 aCasp3/mm$^2$; t-test, p<0.0001), confirming that these two functionally distinct cortical areas are differentially affected by apoptosis during early postnatal period. In layer V, lower rates of cell death are detectable in M1 (sectors a/b: 13.79 ± 2.40 aCasp3/mm$^2$) compared to S1 (sectors c-f: 29.97 ± 2.04 aCasp3/mm$^2$; t-test, p=0.03). Although the cell death density within sectors c/d tends to be higher in layer V than in the other layers, this difference is not significant (two-way ANOVA, post-hoc test p>0.05). The deeper layer VI displays a low amount of apoptotic neurons and does not exhibit any area specificity (two-way ANOVA, sector p>0.05).

The spatial pattern of cell death at P5-7 detected by aCasp3 immunohistochemistry was confirmed by TUNEL staining, which labels fragmented nuclei when cell death process is fully executed (*Gavrieli et al., 1992*). The pattern of TUNEL-positive neurons can be superimposed on the pattern of aCasp3-labeled neurons (*Figure 3B* and *Figure 3—source data 2*; two-way ANOVA, sector p<0.001, detection method p=0.96, n = 3 animals), validating aCasp3 as a reliable marker to identify apoptotic neurons.

For an age-dependent comparison, we analyzed corresponding cortical sections in younger (P1-3; *Figure 3C* and *Figure 3—source data 3*) and older (P9-11; *Figure 3D* and *Figure 3—source data 4*) age groups. In contrast to P5-7 mice, the density of apoptotic neurons detected in both P1-3 and P9-11 mice is very low and does not reveal any sector differences (two-way ANOVA, P1-3: sector p=0.48, n = 8 animals; P9-11: sector p=0.45, n = 9 animals). But, whereas apoptosis similarly affects

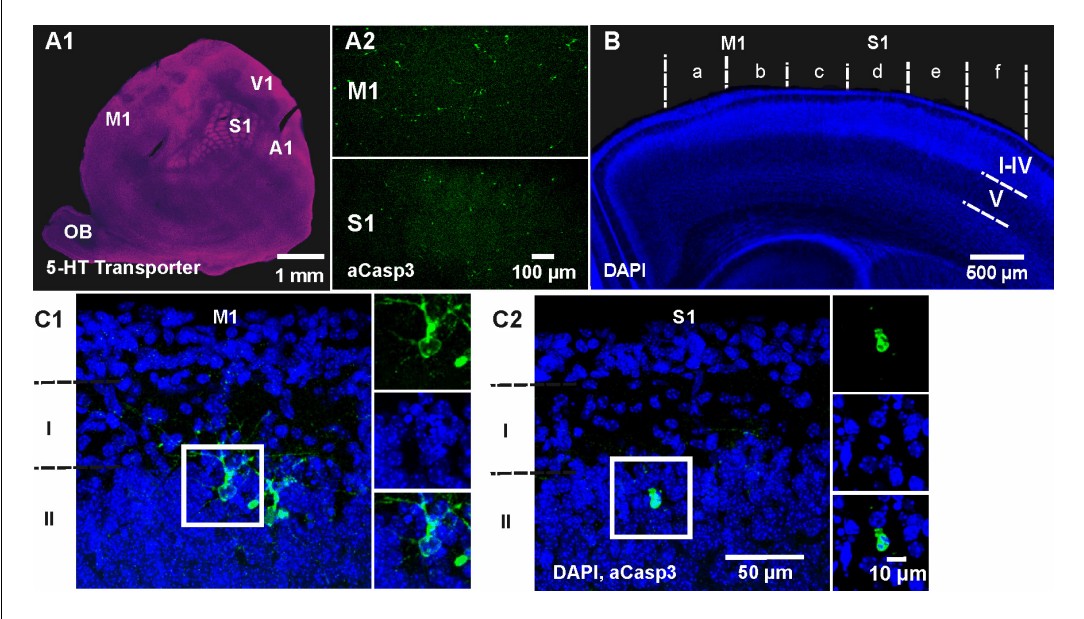

**Figure 2.** Spatiotemporal pattern of neuronal apoptosis in the developing mouse neocortex. (**A1**) Serotonin (5-HT) transporter expression in a flat-mounted, tangential section of P6 brain delineates primary sensory areas. S1, primary somatosensory cortex; M1, primary motor cortex; A1, auditory cortex; V1, visual cortex; OB, olfactory bulb. (**A2**) Co-immunostaining against aCasp3 reveals strong area discrepancies in the density of apoptotic neurons: M1 (upper panel) displays a high density of aCasp3 positive cells compared with S1 (lower panel). (**B**) Representative DAPI staining of a coronal slice at P6 allows the identification of cortical layers and areas. Cortical regions were divided into sectors a to f containing M1 (sectors a and b) and S1 (sectors c to f) for further analysis. Based on DAPI staining, upper layers (layers I-IV), layer V and layer VI were identified. (**C1**) Representative aCasp3-positive neurons in M1 of a coronal slice from a P6 mouse brain. Cell bodies and basal dendrites are located in layers II-IV while apical dendrites project to layer I. Note that aCasp3 allows the detection of dying neurons at an early stage of apoptosis, when aCasp3 signal is not yet translocated into the nucleus and when the nucleus does not display any visible sign of degradation. (**C2**) Representative aCasp3-positive neuron in S1. The aCasp3 signal has translocated into the nucleus (green), the nucleus starts to condense (DAPI, blue) and neurites are not visible anymore.
DOI: https://doi.org/10.7554/eLife.27696.004

all layers in P1-3 mice (layer p=0.08), significantly higher amounts of cell death are detected in layer V than in more superficial layers of older P9-11 mice (layer p<0.0001, post-hoc test sectors b and c < 0.05).

In conclusion, our detailed analysis of the cell death pattern in the developing mouse cortex reveals that the critical time window of cell death takes place between P5 and P7 in all layers of M1 and S1, but distinct layers and cortical areas display striking differences in the extent and distribution of cell death. Particularly in cortical layers I-IV of M1, neurons are more likely to undergo apoptosis as compared to layers I-IV of S1. Further analyses therefore focus on differences in layers I to IV between these two cortical areas.

## Differences in electrical activity of M1 and S1 in vivo parallels area-specific apoptosis

The spatial pattern of apoptosis in the cortex cannot be explained by a different maturation state in M1 versus S1 (*Verney et al., 2000*), but is more likely caused by either area-dependent differences in neuron-intrinsic properties or by area-specific pro-apoptotic versus pro-survival signals. Previous in vitro studies have demonstrated that electrical activity promotes neuronal survival during early development (*Murase et al., 2011*; *Golbs et al., 2011*), indicating that electrical activity functions as a pro-survival factor. Interestingly, previous in vivo recordings of the developing neocortex demonstrated that spontaneous electricalactivity is much more abundant and complex in S1 (*Yang et al., 2009*; *An et al., 2014*) than in M1 (*Yang et al., 2009*; *An et al., 2014*; *Khazipov et al., 2004*). To address the question whether differences in electrical activity can indeed be correlated to different rates of apoptosis in S1 and M1, we performed simultaneous in vivo recordings of spontaneous activity in the upper layers of S1 and M1 of P6 to P7 mice using two 4-shank electrodes (*Figure 4A*

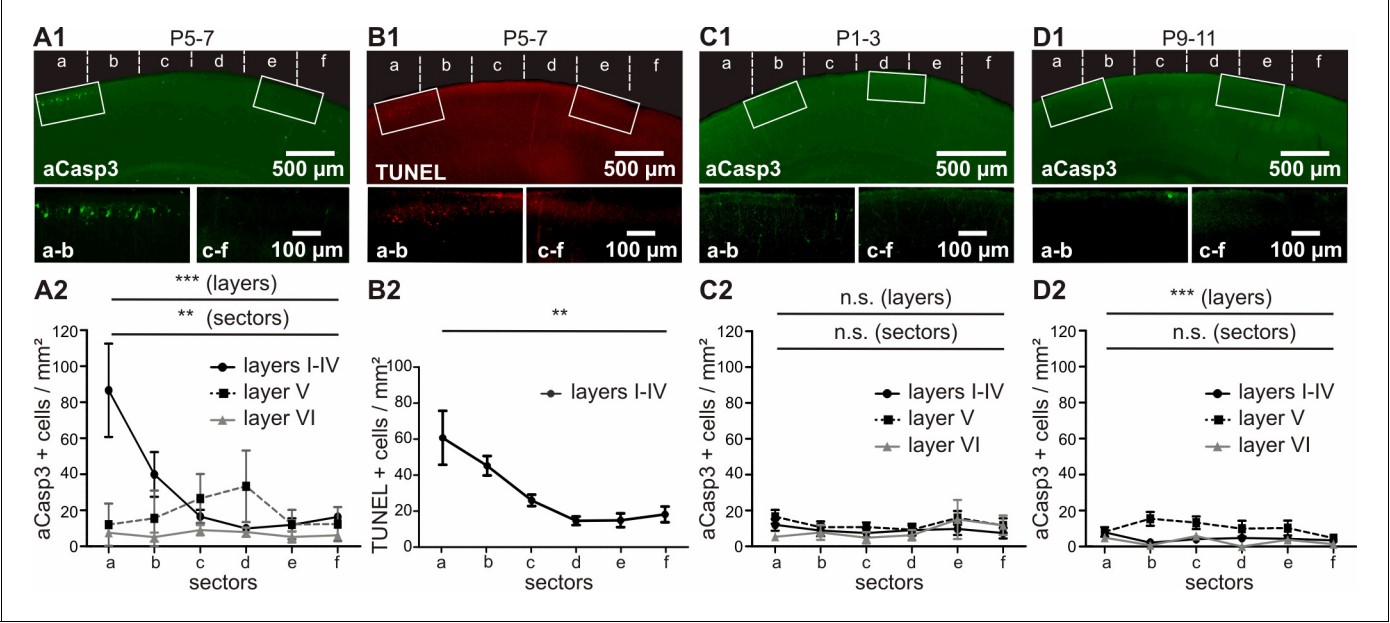

**Figure 3.** In vivo, apoptosis is layer-, sector- and age-dependent. (**A**) In P5-7 mice, cortical layers I-IV display higher cell death densities than deeper layers, with particularly elevated densities of apoptotic cells in M1 compared with S1. (**B**) Comparative labeling of layers I-IV apoptotic neurons with TUNEL staining shows the same pattern as aCasp3 immunostaining. (**C**) In P1-3 mice, the neocortex presents a very low amount of apoptotic cells with no significant difference in density between sectors or layers. (**D**) In P9-11 mice, the number of apoptotic cells is also low, with a lower density in layers I-IV and VI than in layer V and no significant difference between sectors.

DOI: https://doi.org/10.7554/eLife.27696.005

The following source data is available for figure 3:

**Source data 1.** Quantitative analysis of the density of aCasp3-positive cells in layers I-IV, V and VI of P5-7 mouse neocortex.
DOI: https://doi.org/10.7554/eLife.27696.006

**Source data 2.** Quantitative analysis of the density of TUNEL-positive cells in layers I-IV of P5-7 mice neocortex.
DOI: https://doi.org/10.7554/eLife.27696.007

**Source data 3.** Quantitative analysis of the density of apoptotic cells in layers I-IV, V and VI of P1-3 mouse neocortex.
DOI: https://doi.org/10.7554/eLife.27696.008

**Source data 4.** Quantitative analysis of the density of apoptotic cells in layers I-IV, V and VI of P9-11 mouse neocortex.
DOI: https://doi.org/10.7554/eLife.27696.009

*and B*). As previously reported for early neocortical networks in vivo (*Yang et al., 2009*; *An et al., 2014*; *Khazipov et al., 2004*), both cortical regions display intermittent network oscillatory activity (*Figure 4C*). Comparative analysis of the in vivo network activity recorded in the two cortical areas demonstrates that S1 activity is characterized by a significantly higher spectral power in the delta to gamma range than M1 activity (*Figure 4D1 and D2* and *Figure 4—source data 1*; two-way ANOVA, cortical area p<0.0001; frequency p<0.0001; n = 5 animals). A detailed analysis of this activity further reveals that in S1, the overall number of troughs (*Figure 4D3* and *Figure 4—source data 1*; 156 ± 27.8 min$^{-1}$, n = 5 animals) and the median trough frequency (*Figure 4—figure supplement 1*; 1.43 Hz, n = 2333 events) is significantly higher than the overall number of troughs (63 ± 20 min$^{-1}$; Mann Whitney test, p=0.0317) and median trough frequency (0.82 Hz; n = 944 events; Mann Whitney test, p<0.001) in M1. To verify that the area-dependent level of activity is not due to a differential effect of urethane between M1 and S1, we next recorded the activity in the upper layers of S1 and M1 in non-anesthetized animals. Also under this condition, the spectral power of S1 in vivo activity is significantly higher than the spectral power of M1 in vivo activity (*Figure 4E1 and E2* and *Figure 4—source data 2*; two-way ANOVA, cortical area p=0.0015, n = 5 animals). Furthermore, the trough rate in S1 (*Figure 4E3* and *Figure 4—source data 2*; 159.7 ± 23.6 min$^{-1}$) is significantly higher than the trough rate in M1 (51.9 ± 23.4 min$^{-1}$; Mann Whitney test p=0.03). Thus, also in non-anesthetized animal, the activity levels in S1 are significantly higher than that in M1.

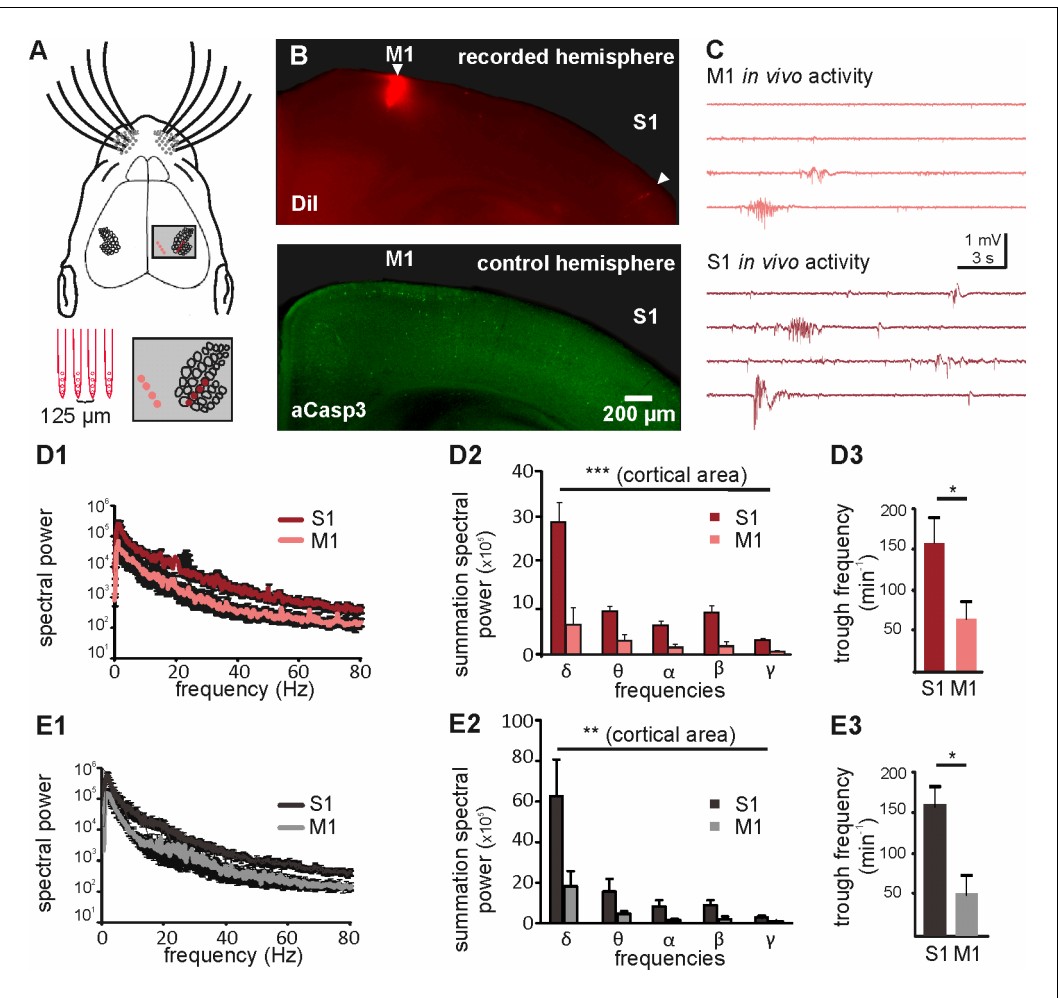

**Figure 4.** Different levels of spontaneous activity in somatosensory and motor cortical areas in urethane-anesthetized and non-anesthetized mice in vivo. (A) Location of the 4-shank recording electrode in M1 and S1 (upper panel and close-up in grey-shaded area) and schematic illustration of the electrodes (lower left panel). (B) Tracks of DiI marked electrodes illustrate recording positions in M1 and S1 in a coronal slice (upper panel, filled arrows). In the opposite control hemisphere from the same slice (lower panel), aCasp3 expression reveals that M1 recording site corresponds to a region of high rate of apoptosis and S1 recording site to a region of low apoptosis. (C) Representative in vivo local field potential (LFP) recordings performed simultaneously in M1 and S1 of urethane-anesthetized mice. (D) In urethane-anesthetized mice, (D1) the averaged spectral power from continuous LFP recordings, (D2) the summation of LFP spectral power and (D3) the detected trough rate are significantly higher in S1 than in M1. (E) Also in non-anesthetized mice, (E1) the averaged spectral power from continuous LFP recordings, (E2) the summation of LFP spectral power and (E3) the trough rate are significantly higher in S1 than in M1.

DOI: https://doi.org/10.7554/eLife.27696.010

The following source data and figure supplement are available for figure 4:

**Source data 1.** Analysis of *in vivo* recordings in S1 and M1 from urethane-anesthetized mice.
DOI: https://doi.org/10.7554/eLife.27696.012
**Source data 2.** Analysis of *in vivo* recordings in S1 and M1 from non-anesthetized mice.
DOI: https://doi.org/10.7554/eLife.27696.013
**Figure supplement 1.** Analysis of in vivo recordings from S1 and M1.
DOI: https://doi.org/10.7554/eLife.27696.011

In summary, these data demonstrate that motor and somatosensory cortical areas of immature mice display different levels of spontaneous activity that may underlie the observed area-dependent differences in the rate of apoptosis.

## Increased electrical activity rescues cortical neurons from cell death in an age-dependent manner

In order to address the question whether differences in electrical activity between M1 and S1 are causally related to the different rates of apoptosis between these two cortical areas, we next used in vitro slice preparations that allow a better control of electrical activity.

Before proceeding to pharmacological modulation of ongoing electrical activity in our acute cortical sections containing both S1 and M1 (*Figure 2B*), we validated the preparation as suitable to study area-dependent developmental apoptosis. Following dissection, slices were kept in vitro under interface conditions in ACSF and extracellular evoked field postsynaptic responses (fEPSP) were recorded every hour to validate the slice's viability (*Figure 5—figure supplement 1A*). Subsequent immunhistochemical analysis revealed that these in vitro slices display a pattern of neuronal apoptosis similar to the in vivo situation – that is with a high cell death density in cortical layers I-IV in M1 and significantly sparser aCasp3-positive neurons in the more lateral S1 sectors (*Figure 5A2 and A3*; one-way ANOVA, $R^2$=0.34, p<0.0001, n = 7 hemispheres from seven animals). But in comparison to the in vivo situation, the overall rate of cell death is increased by about 4-fold (in vivo: 34.39 $\pm$ 14.44 aCasp3/mm$^2$; in vitro: 132.67 + 39.2 aCasp3/mm$^2$, t-test = 0.040). Thus, this in vitro model preserves the typical area-dependent pattern of apoptosis observed in vivo, but exhibits an exacerbated apoptotic process.

We next asked whether inducing high levels of electrical activity by bath application of the voltage-dependent $K^+$ channel blocker 4-aminopyridine (4-AP) in combination with depletion of extracellular $Mg^{2+}$ (*Kilb et al., 2006*) could rescue immature cortical neurons from apoptosis. Under these conditions, recorded extracellular activity consists of repetitive discharges occurring at a frequency of 0.3 $\pm$ 0.04 Hz and an average amplitude of 106 $\pm$ 16 µV (*Figure 5A1*; n = 6 slices from six animals). This pharmacological-induced activity spreads over the whole cortex as confirmed by planar multi-electrode array (MEA) recordings (data not shown). Subsequent analysis of these slices shows significantly lower numbers of apoptotic neurons compared to untreated slices (*Figure 5A3* and *Figure 5—source data 1*; two-way ANOVA, sector p<0.0001, electrical activity p<0.0001, n = 6 slices from six animals).

To exclude the possibility that lower rates of apoptosis under high electrical activity conditions only reflect an increased apoptosis process at the early phase of the experiment with subsequent loss of neurons, slices were analyzed 3 hr following dissection. At this earlier time point, rates of apoptosis are already decreased in slices exposed to 4-AP/0 $Mg^{2+}$ for two hours compared to control conditions (*Figure 5—figure supplement 1B*; two-way ANOVA, sector p=0.001, electrical activity p=0.02).

To prove that the pro-survival effect of the 4-AP/0 $Mg^{2+}$ treatment does not depend on the pharmacological model, but is rather caused by increased spontaneous electrical activity per se, we additionally investigated the effect of disinhibition-induced activity on apoptotic rates by perfusing the slices with 3 µM of the GABA$_A$ receptor antagonist gabazine in low $Mg^{2+}$ conditions. Under this condition, repetitive single discharges occur at a frequency of 0.24 $\pm$ 0.01 Hz with an average amplitude of 358.5 $\pm$ 82 µV (*Figure 5B1*, n = 6 slices from six animals). The elevated levels of activity induced by this second pharmacological model likewise result in a strong reduction in the rate of apoptosis in comparison to untreated control slices (*Figure 5B* and *Figure 5—source data 2*; two-way ANOVA, sector p=0.009, electrical activity p=0.001, n = 6 untreated hemispheres from six animals). These data indicate that in vitro, increasing spontaneous activity for two hours are sufficient to significantly lower the amount of neurons undergoing cell death in the immature cortex.

Additional control experiments to investigate whether these effects are restricted to early developmental stages demonstrate that the density of apoptotic cells in adult (P27) cortical neurons is very low in comparison to P5-7 mice (*Figure 5A3* and *Figure 5—source data 1*; two-way ANOVA, age p<0.0001, n = 6 hemispheres from three animals) and homogenously distributed throughout all cortical areas analyzed (sectors p=0.30). Interictal and ictal-like activity induced by 4-AP/0 $Mg^{2+}$ treatment (*Figure 5—figure supplement 1C*, n = 6 hemispheres from three animals) is not correlated to enhanced apoptotic rates in these slices (electrical activity p=0.27). These results

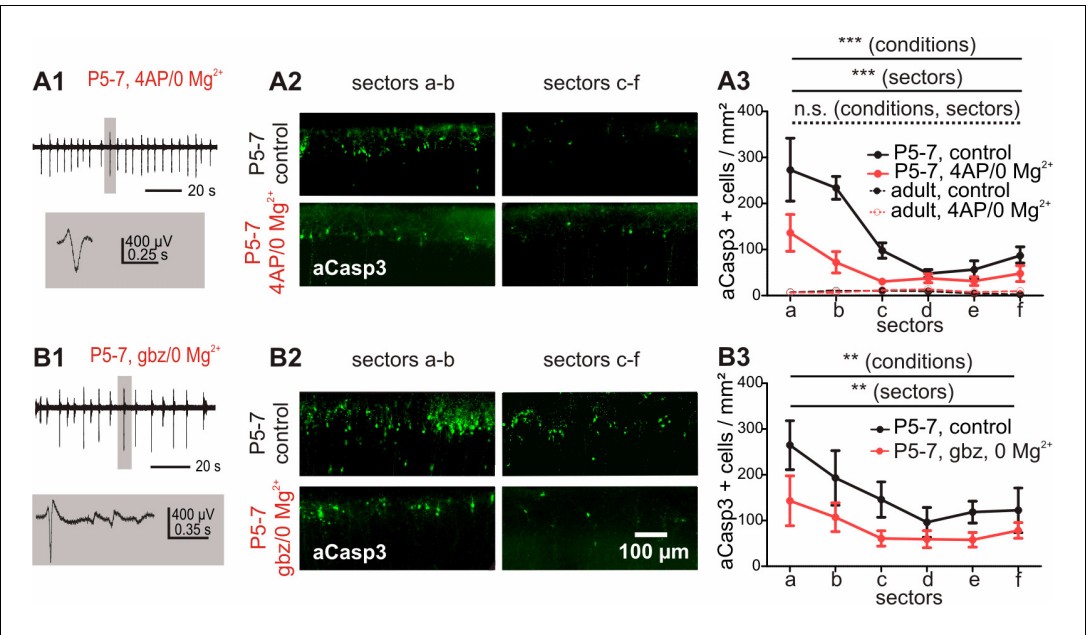

**Figure 5.** Increased electrical activity in vitro causes a reduction in apoptosis. (**A1**) Typical LFP activity in cortical slices from P5-7 mice induced by 4AP/0 $Mg^{2+}$ ACSF. (**A2 and A3**) Under control conditions, in vitro cortical slices (black solid line) present the same area-specific pattern of apoptosis as in vivo although rates of apoptosis are increased. Bath application of 4AP/0 $Mg^{2+}$ ACSF reduces the extent of cell death in vitro (red solid line). Control experiments in adult cortical slices (black dotted lines) reveal a lower density of apoptotic neurons which is insensitive to 4AP/0 $Mg^{2+}$ treatment (red dotted lines). (**B1**) Application of 3 µM gabazine (gbz)/0 $Mg^{2+}$ ACSF also increases the spontaneous electrical activity in coronal slices from P5-7 mice and significantly reduces the amount of apoptotic neurons (**B2 and B3**).

DOI: https://doi.org/10.7554/eLife.27696.014

The following source data and figure supplements are available for figure 5:

**Source data 1.** Quantitative analysis of the density of aCasp3-positive cells in layers I-IV of acute slices from P5-7 or adult mouse neocortex perfused with control or 4-AP/no $Mg^{2+}$ ACSF.

DOI: https://doi.org/10.7554/eLife.27696.017

**Source data 2.** Quantitative analysis of the density of aCasp3-positive cells in layers I-IV of acute slices from P5-7 mouse neocortex perfused with control or gabazine/no $Mg^{2+}$ ACSF.

DOI: https://doi.org/10.7554/eLife.27696.018

**Figure supplement 1.** Field potential recordings in coronal cortical slices under control and high frequency activity conditions.

DOI: https://doi.org/10.7554/eLife.27696.015

**Figure supplement 2.** The area-specific pattern of apoptosis disappears in a cortical preparation with homogenous electrical activity.

DOI: https://doi.org/10.7554/eLife.27696.016

demonstrate that both the high occurrence of neuronal apoptosis and the pro-survival effect of increased activity are hallmark features of immature cortical networks.

Interface in vitro slice preparations only allow investigation for a restricted period. To study whether a prolonged application of identical electrical activity levels are sufficient to match apoptotic rates in S1 and M1, we used organotypic cultures. Organotypic neocortical slices cultivated from newborn mice typically display spontaneous recurrent waves of activity propagating over the whole cortex (*Conhaim et al., 2011*; *Heck et al., 2008*). In line with our hypothesis, quantification of aCasp3-positive neurons in layers I to IV after 4 days in vitro (corresponding to P5-7) does not reveal any significant difference in the rate of apoptosis between cortical sectors covering M1 and S1 (*Figure 5—figure supplement 2*; one-way ANOVA, sector p=0.74, $R^2$=0.07, n = 7 slices from seven animals). Hence, regional differences in rates of apoptosis are abolished in the presence of equivalent spontaneous electrical activity levels. Upon silencing of electrical network by 1 µM TTX, the overall

rate of apoptosis significantly increases (*Figure 5—figure supplement 2*; two-way ANOVA, condition p=0.0001, n = 3 slices from three animals), demonstrating that an activity-dependent process underlies these observations.

In conclusion, these results demonstrate that elevated electrical activity exerts a strong pro-survival signal on immature cortical neurons during the critical period of cell death and diminishes the area-specific profile of apoptosis, indicating that the spatial pattern does not rely only on the area identity of cortical neurons.

## Replay of in vivo S1 activity has a stronger anti-apoptotic effect than replay of in vivo M1 activity

In order to substantiate our hypothesis that different levels of activity control area-specific rates of apoptosis, we next directly investigated whether the activity pattern recorded in S1 represents a stronger pro-survival effect than the M1 in vivo activity. For this purpose, we translated the in vivo patterns of activity recorded in M1 and S1 into protocols for in vitro electrical stimulation (*Figure 6—figure supplement 1*). These in vitro stimulus protocols were replayed via bipolar stimulation electrodes located in layers V-VI of the M1 region of coronal slice preparations (*Figure 6A*). The replay pattern of M1 activity consisted of 24 ± 3.1 electrical stimuli/min (n = 2 stimulus protocols) and the replay pattern of S1 activity had a significantly higher rate of 125 ± 15.2 stimuli/min (n = 2 stimulus protocols). Both stimulation protocols and the resulting activity resembled the temporal distribution of events in vivo (*Figure 6A2* and *Figure 6—figure supplement 1*). As expected from our previous results, replay of in vivo activity in layers V-VI of the motor cortex results in a significant reduction of aCasp3-expressing neurons in the stimulated hemisphere when compared to the unstimulated, contralateral hemisphere (*Figure 6C* and *Figure 6—source data 1*; two-way ANOVA, sector p<0.0001, electrical activity p=0.0008, n = 6 slices from six animals). In line with our hypothesis, replay of S1 activity has a significantly higher pro-survival effect than replayed M1 activity (*Figure 6D* and *Figure 6—source data 2*; two-way ANOVA, stimulation pattern p=0.03, area p=0.63).

In summary, these results provide evidence that the different activity levels in M1 and S1 can be causal for the observed differences in area-specific rates of apoptosis.

## Altered electrical activity affects rates of apoptosis and reduces regional differences in vivo

We next examined whether increasing the level of ongoing electrical activity in P6 mice in vivo could attenuate the amount of cortical neurons undergoing apoptosis in regions of naturally high apoptosis. Consistent with previous work in P6 rats (*Puskarjov et al., 2015*), a single intraperitoneal injection of kainate (2 mg/kg) induces tonic seizures in P5-7 mice within about 40 min. Quantification of apoptosis in M1 and S1 regions of kainate- versus saline-injected mice demonstrates that also in vivo, modifying electrical activity levels affects the susceptibility of immature neurons to undergo apoptosis (*Figure 7A* and *Figure 7—source data 1*; two-way ANOVA, sector p<0.0001, condition p=0.005). Whereas in the somatosensory cortex, rates of apoptosis are not significantly reduced upon induction of epileptiform activity (control: 14.09 ± 2.49 aCasp3 positive cells/mm$^2$, n = 7 mice; kainate: 9.28 ± 1.57 aCasp3 positive cells/mm$^2$, n = 9 mice, post-hoc test p>0.05), levels of apoptosis in the primary motor cortex are significantly lower in kainate-injected mice (30.25 ± 6.16 aCasp3 positive cells/mm$^2$) than in saline-injected mice (52.03 ± 6.00 aCasp3 positive cells/mm$^2$, post-hoc test p<0.01). This result demonstrates that also in vivo, high levels of activity significantly lower the incidence of neuronal apoptosis, especially in a cortical area that encounters physiologically low levels of electrical activity, further supporting our hypothesis that neuronal activity controls the area-specific pattern of apoptosis.

To substantiate these in vivo findings, we next reduced the activity in S1 by sensory deafferentation. A substantial portion of electrical activity in S1 is generated in the sensory periphery (*Yang et al., 2009*; *Minlebaev et al., 2007*; *Yang et al., 2016*) and this portion may even be increased under physiologically relevant behavioral condition like nursing (*Sullivan et al., 2003*; *Akhmetshina et al., 2016*). Hence, deprivation of sensory inputs by whisker clipping and simultaneous hourly lidocaine (2% gel) application to the whisker pad massively reduces the level of electrical activity in S1 (*Yang et al., 2009*). Quantification of aCasp3-labeled cells shows that already within 5 hr, whisker deafferentation leads to an increased density of apoptotic neurons in the

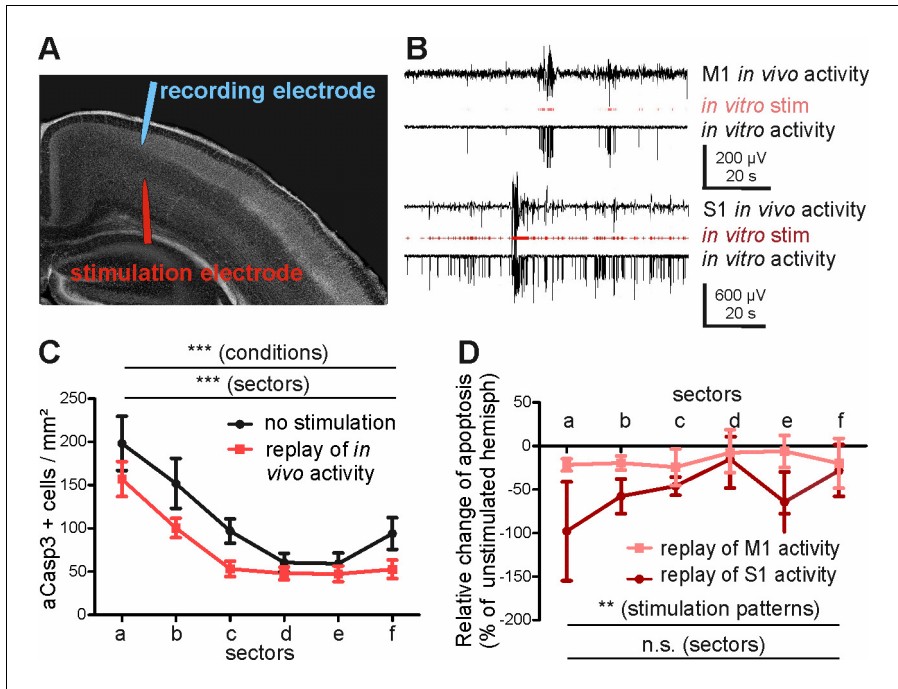

**Figure 6.** Replay of in vivo S1 activity has a stronger anti-apoptotic effect than replay of in vivo M1 activity. (**A**) DAPI image of a slice schematically illustrating the position of the stimulation electrode in layer V of M1 and of the recording electrode in layers II-III of M1. (**B**) Typical LFP recordings illustrating in vivo M1 activity (upper trace), the stimulus sequence extracted from the in vivo recording (red trace) and the resulting LFP recording in an acute slice using this stimulus sequence (lower trace) for either M1 (upper panel) or S1 (lower panel) activity pattern. (**C**) Statistical analysis demonstrating that repetitive electrical stimulation using replay of S1 and M1 in vivo activity decreases the density of apoptotic neurons in the stimulated hemisphere (red symbols/lines) compared to unstimulated contralateral hemisphere (black symbols/lines). (**D**) Statistical analysis of the decrease in the number of aCasp3-positive neurons related to the unstimulated hemisphere. The relative reduction in number of aCasp3-labeled neurons is higher after replay of S1 in vivo pattern than after replay of M1 in vivo pattern .
DOI: https://doi.org/10.7554/eLife.27696.019

The following source data and figure supplement are available for figure 6:

**Source data 1.** Quantitative analysis of the density of aCasp3-positive cells in layers I-IV of acute slices from P5-7 mouse neocortex with and without replay of S1 and M1 in vivo activity.
DOI: https://doi.org/10.7554/eLife.27696.021

**Source data 2.** Quantification of decrease in the number of aCasp3-positive neurons following electrical stimulation (related to the unstimulated hemisphere).
DOI: https://doi.org/10.7554/eLife.27696.022

**Figure supplement 1.** Generation of in vitro stimulus protocols replaying S1 and M1 in vivo activities.
DOI: https://doi.org/10.7554/eLife.27696.020

corresponding contralateral barrel field (*Figure 7B* and *Figure 7—source data 2*; 18.74 ± 1.88 aCasp3 positive cells/mm$^2$, n = 10 mice) as compared to the somatosensory cortex of the ipsilateral hemisphere receiving non-obstructed whisker afferents (13.77 ± 1.41 aCasp3 positive cells/mm$^2$; t-test, p=0.038). These results provide direct evidence that periphery-driven electrical activity in S1 provides an anti-apoptotic signal that contributes to the area-specific differences in rates of apoptosis.

In summary, our data show for the first time that electrical activity regulates cell death in the developing cortex in a cortical area-dependent manner and thus represents an important factor for the functional and structural maturation of the brain.

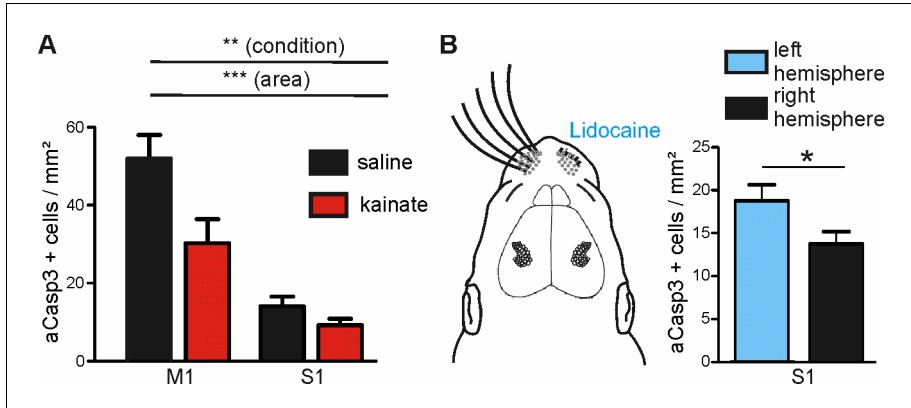

**Figure 7.** Modulation of spontaneous activity in vivo affects neuronal apoptosis in the developing cortex. (**A**) Increasing spontaneous electrical activity for 5 hr by kainate injection significantly decreases rate of apoptosis in M1 whereas it does not affect neuronal apoptosis in S1. (**B**) Left panel: Schematic illustration of sensory deprivation by unilateral clipping of the whiskers and local lidocaine application into the whisker pad for 5 hr.
Right panel: Unilateral whisker deafferentation leads to increased amounts of apoptosis in the corresponding contralateral barrel field in comparison to the ipsilateral hemisphere receiving intact whisker afferents.
DOI: https://doi.org/10.7554/eLife.27696.023

The following source data is available for figure 7:

**Source data 1.** Quantitative analysis of the density of aCasp3-positive cells in M1 and S1 areas of saline- or kainate-injected P5-7 mice.
DOI: https://doi.org/10.7554/eLife.27696.024

**Source data 2.** Quantitative analysis of the density of aCasp3-positive cells in S1 area of P5-7 mice.
DOI: https://doi.org/10.7554/eLife.27696.025

## Discussion

While size, complexity and processing capacity of the cerebral cortex have incredibly expanded during mammalian evolution, apoptosis (*Oppenheim, 1991*) and early spontaneous synchronous activity (*Khazipov and Luhmann, 2006*), two temporally overlapping hallmark features of immature cortical networks, are phylogenetically conserved. It is therefore reasonable to hypothesize that both processes - electrical activity and programmed cell death - are essential elements which might not only co-occur, but mutually interact to ensure the proper development of the cortex. The experiments provided in this study support this hypothesis as they demonstrate for the first time that (i) M1 exhibits low neuronal activity and a high number of apoptotic neurons while S1, with a higher rate of spontaneous activity, shows less apoptotic neurons, (ii) enhancing neuronal activity in vitro and in vivo reduces the rate of apoptosis in M1, (ii) in vitro replay of S1 activity recorded in vivo to M1 reduces apoptosis in this area, and (iv) suppressing sensory-driven activity in vivo enhances the number of apoptotic neurons in S1. We conclude from these results that neuronal activity is a major determinant of the region-dependent rate of apoptosis in the developing neocortex.

In accordance with previous reports (*Ferrer et al., 1994*; *Southwell et al., 2012*; *Nikolić et al., 2013*), the present study shows that the immature cortex displays higher rates of apoptosis than the adult cortex. We additionally demonstrate that during the first postnatal week, functionally diverse neocortical areas reveal distinct levels of neuronal apoptosis. While a mediolateral apoptosis profile of neocortical neurons comparable to the M1 versus S1 difference reported in the present study has been shown previously (*Ikonomidou et al., 1999*; *Verney et al., 2000*; *Mosley et al., 2017*), a direct relation of apoptotic densities to functional neocortical areas has, to our knowledge, not been reported before. Our data are an important extension of previous observations demonstrating neuron-specific (*Derer and Derer, 1990*; *Del Río et al., 1996*; *Robertson et al., 2000*; *Blanquie et al., 2017b*) and brain-region specific (*Ikonomidou et al., 1999*; *Verney et al., 2000*; *Mosley et al., 2017*; *Ahern et al., 2013*) rates of cell death, and thus underline the important role of apoptosis for the establishment of functionally distinct neural circuits during early development (*Ferrer et al., 1994*; *Southwell et al., 2012*; *Nikolić et al., 2013*).

It has been demonstrated in vitro that the rate of neuronal apoptosis at early developmental stages is to a large extent negatively regulated by electrical/synaptic activity (*Golbs et al., 2011*). In line with these in vitro results, the present study confirms that an elevation of neuronal activity by pro-convulsive conditions diminishes the rates of apoptosis in the immature neocortex under in vitro and in vivo conditions. We further show that already two hours of elevated electrical activity are sufficient to significantly decrease the number of apoptotic cortical neurons. As immature brains are highly susceptible to acute changes in electrical activity (*Ben-Ari and Holmes, 2006*), this stresses the clinical relevance of the presented data. Interestingly, neocortical slices reveal a higher number of apoptotic neurons compared to the same neocortical areas in vivo. This observation could be due to the lower rate of spontaneous electrical activity in acute slice preparations. Indeed, increasing neuronal activity by pharmacological or electrical stimulation reduces the number of apoptotic neurons in vitro. Interestingly, also under in vivo conditions, the rate of apoptosis is reduced upon application of the pro-convulsive agent kainate. This effect is only significant for M1, but not for S1, either because the smaller number of apoptotic neurons in S1 impedes the analysis of such rescue experiments, or because differences in the electrical activity between S1 and M1 influence the amount of additional spikes introduced by convulsive conditions.

We observe a higher rate of electrical activity in S1 than M1, as quantified by the higher number of events as well as by the higher median discharge frequency, indicating a higher firing frequency during bursts of activity. This observation is in line with several studies reporting a higher occurrence and greater complexity of spontaneous activity in S1 compared to M1 (*Yang et al., 2009*; *An et al., 2014*; *Khazipov et al., 2004*). The negative correlation observed between activity levels and rates of neuronal apoptosis in S1 versus M1 suggests that differences in spontaneous activity levels underlie the distinct rates of apoptosis between the two neocortical areas. To provide causal evidence for this hypothesis, we first demonstrate in vitro that replay of S1 activity to M1 significantly reduces the number of apoptotic neurons, while replay of M1 activity has a significantly lower pro-survival effect. In addition, we show in vivo that blocking peripheral activity arising from the mystacial whiskers, which decreases the frequency of spindle bursts and gamma oscillations in S1 by about 50% (*Yang et al., 2009*), significantly reduces the rate of apoptosis in the corresponding S1 barrel cortex. In summary, these results indicate that electrical activity and levels of apoptosis are negatively correlated and provide strong evidence that regional differences in electrical activity are indeed causal for the different levels of apoptosis in functionally distinct neocortical areas. One additional intriguing observation of the present study is that organotypic slices, in which the electrical activity is synchronized over the whole preparation, do not display any regional differences in apoptotic rates. Since in this preparation blockade of all electrical activity with TTX also does not uncover regional differences, it is tempting to postulate that electrical activity per se provide the major regionalization cue regulating specific rates of apoptosis. While the effect of elevated neuronal activity dramatically reduces the number of neurons undergoing apoptosis already after 4 hr, the effect of chronic activity blockade seems to be more moderate. In vivo, the absence of synaptic transmission leads to a dramatic apoptotic response directly after neurons have assembled (*Verhage et al., 2000*). We therefore hypothesize that the survival of cortical neurons in silenced organotypic cultures is not due to an alternative pro-survival pathway but rather to the presence of pro-survival factors present within the culture medium. However, which factor promotes the survival of which neurons under these conditions remains an open question.

We also observe differences in the density of apoptotic neurons between layers within one cortical area. This finding is in line with a previous report demonstrating that layers II to IV show more apoptotic neurons than layers V and VI in the first postnatal week (*Heumann and Leuba, 1983*). However we find this pattern only in M1. In S1, the density of apoptotic neurons in layer V tends to be higher than in layers I-IV. Surprisingly, in vivo recordings in the M1 region of P6 mice across all neocortical layers do not show a negative correlation between activity levels and rates of apoptosis (unpublished data). Whether M1 neurons present a layer-dependent intrinsic sensitivity to electrical activity as a pro-survival factor or whether other factors rule the cell death of deep layer neurons remains unknown. In the future, investigating the correlation between activity levels and rates of apoptosis across layers, that is in regions of different cellular composition, will help uncover cell type-specific differences in the regulation of neuronal apoptosis (*Blanquie et al., 2017a*).

Lower layer VI contains the early-generated subplate neurons (*Hoerder-Suabedissen and Molnár, 2013*; *Kanold and Luhmann, 2010*; *Higashi et al., 2005*). It has been suggested that a

proportion of these neurons is eliminated by apoptosis during the first postnatal weeks (*Price et al., 1997*). Interestingly, we found a low density aCasp3-labeled neurons in the layer VI. We attribute this low cell death density to two reasons. First, the subplate layer represents a very thin layer of about 50 μm-width located between the white matter and layer VI-a (*Hoerder-Suabedissen and Molnár, 2013*). Because, apoptotic neurons are cleared away by macrophages with 2.5 hr (*Thomaidou et al., 1997*), the chance to stain apoptotic neurons in this layer is very low. Second, recent reports have demonstrated that only a fraction of subplate neurons is eliminated by apoptosis, while other subplate neurons such as Nurr1 or Lpar1-GFP-labeled neurons survive beyond the postnatal period (*Hoerder-Suabedissen and Molnár, 2013*). In line with the existence of a substantial fraction of surviving subplate neurons, it has been recently shown that the layer VI-b consists of persistent neurons from the subplate layer (*Marx et al., 2017*).

In the immature brain, including that of humans, a variety of intrinsically-generated and sensory-evoked activity patterns can be observed in different cortical areas (*An et al., 2014*; *Milh et al., 2007*; *Yang et al., 2009*; *Khazipov et al., 2004*; *Khazipov and Luhmann, 2006*; *Kilb et al., 2011*). The distinct levels of spontaneous neuronal activity in the immature sensory, motor or association cortex have been frequently attributed to the development of adequate connectivity (*Chen and Ghosh, 2005*; *Luhmann et al., 2016*; *Spitzer, 2012*; *Hartung et al., 2016*). The results of the present study demonstrate that a direct influence of sensory-driven and spontaneous neuronal activity on the size and probably also the composition of the cellular pool underlying neuronal networks must also be considered as an important element during brain formation.

Human brains display activity patterns similar to the rodent ones (*Khazipov and Luhmann, 2006*; *Vanhatalo and Kaila, 2006*). For instance the electroencephalogram of preterm babies exhibits at an equivalent developmental period a local rhythmic activity in a similar frequency range than spindle bursts: the so-called delta brushes (*Milh et al., 2007*). These human preterm activity patterns most probably also modulate neuronal apoptosis in the developing human brain, which points to an additional mechanism by which inadequate activity can lead to persistent changes in the nervous system. As revealed by previous animal studies, subsequent modification of the number of cortical neurons results in pathophysiological outcomes such as autistic-like behavior (*Fang et al., 2014*; *Nakamura et al., 2016*). In line with behavioral deficits as a consequence of changes in the neuronal population size, a large fraction of children suffering from autistic spectrum disorders shows an elevated number of neocortical neurons (*Courchesne et al., 2011*; *Casanova et al., 2006*). While it has been proposed that mostly enhanced neurogenesis causes this autism-related increase in cerebral volume (*Kaushik and Zarbalis, 2016*), we propose that activity-dependent alterations in apoptosis must be also considered as possible cause. Further, a recent study reported that sensory whisker deprivation during the first postnatal days, that is through the period covering the critical period of cell death, induces not only the expected deficits in tactile sensation but also exacerbated fear and anxiety-related responses as well as impaired social behavior (*Soumiya et al., 2016*). In conclusion we demonstrate that neuronal electrical activity at pre- and peri-natal stages, besides affecting neuronal proliferation, migration or integration (*De Marco García et al., 2015*, *2011*; *Weissman et al., 2004*; *López-Bendito and Molnár, 2003*), also directly influences apoptosis. This implies on the one hand that physiological activity patterns are essential for the establishment and refinement of functional circuits in the cortex; and that drugs or pathophysiological conditions interfering with early activity and cell death likely contribute to many neuropsychiatric diseases (*Brown et al., 2005*; *van Diessen et al., 2015*; *Ben-Ari, 2015*; *Silva et al., 2005*).

## Materials and methods

### Animals

All experiments were conducted in accordance with National and European (86/609/EEC) laws for the use of animals in research and were approved by the local ethical committee (Landesuntersuchungsamt Rheinland-Pfalz 23.177–07/G 10-1-010). Offspring from timed-pregnant C57BL/6NRj wild-type mice (Janvier labs) were used for all experiments. In a subset of experiments, GAD67-GFP knock-in mice were used, in which an enhanced GFP was targeted to the *Gad67* locus that encodes GAD67 (*Tamamaki et al., 2003*).

## Slice preparation

For immunhistochemical analysis of coronal sections, P1 to P11 mice were decapitated, brains were removed from the skull and quickly submersed into 4% paraformaldehyde (PFA) at 4°C overnight. Brains were then cryoprotected in 30% sucrose and 50 µm coronal sections were cut on a cryostat (Leica, Wetzlar, Germany). Tangential sections were prepared as follows: brains were dissected and cut along the midline. Each hemisphere was tangentially sectioned and then flattened between glass slides with 1 mm spacers. Sections were post-fixed overnight in 4% PFA, cryoprotected and 200 µm sections were cut on a cryostat.

For electrophysiological recordings, acute coronal slices were prepared as follows: mice were killed by decapitation, brains were quickly removed and immersed in ice-cold standard artificial cerebrospinal fluid (ACSF). All solutions were equilibrated with 95% $O_2$/5% $CO_2$ at least 1 hr before use and standard ACSF consisted of 126 mM NaCl, 26 mM NaHCO3, 1.25 mM $NaH_2PO_5$, 1 mM $MgCl_2$, 2 mM $CaCl_2$, 2.5 mM KCl, 10 mM glucose. Coronal brain slices of 500 µm thickness were cut on a Vibratome (Microm HM 650V, Thermo Fisher Scientific, Germany). Slices at 4–5 mm distance from the most rostral section (*Paxinos et al., 2007*) were used for subsequent electrophysiological recordings. Afterwards, slices were fixed in 4% PFA at 4°C overnight and 50 µm-thick subsections were cut after cryoprotection on a cryostat.

Cortical organotypic slice cultures were prepared from postnatal day (P)2 to 3 mice as described previously (*Stoppini et al., 1991*; *Heck et al., 2008*). In brief, mice were killed by decapitation, brains were quickly removed and transferred into ice-cold culture medium with the following composition: MEM (Gibco) supplemented with 25% horse serum and HBSS$^{+/+}$, 1 mM Glutamax, 6 mg/mL of glucose and pH adjusted to 7.2. Neocortical hemispheres were isolated from subcortical structures and 350 µm coronal slices were sectioned with a tissue chopper (McIlwain, Mickle Laboratory Engineering, Surrey, UK). Cortical slices were transferred onto Millicell membranes (Sigma-Aldrich, Steinheim, Germany) placed into 35 mm Petri dishes as support for explants. Slices were kept at 37°C in humidified 95% air and 5% $CO_2$. Culture medium was renewed the following day and 1 µM tetrodotoxin citrate (TTX) (Tocris, Bristol, UK) or water for control was added. After 4 DIV, slices were fixed in 4% PFA for 30 min (min) and directly processed for immunohistochemistry.

## Extracellular field potential recordings and analysis

Extracellular field potential recordings were performed as described previously (*Kilb et al., 2006*). Briefly, after preparation, slices were directly transferred into an interface-type recording chamber where they were continuously perfused with control ACSF at a rate of 1–2 ml/min at 31 ± 1°C. Slices were allowed to recover for at least 1 hr under these conditions. $MgCl_2$ was removed from solution for 0 $Mg^{2+}$ solutions and either 20 µM 4-AP (Sigma-Aldrich) or 3 µM gabazine (SR 95531 hydrobromide; Tocris) dissolved in dimethylsulfoxide (DMSO, Sigma) were added. Slices were perfused with either ACSF for 4 hr before being fixed and processed for immunostaining. The DMSO concentration of the final solution never exceeded 0.2%. Extracellular field potentials were recorded with tungsten microelectrodes (impedance 4–5 MOhm; FHC, Bowdoinham, ME, USA) in layer II of the auditory cortex. Signals were amplified by a purpose built amplifier, low-pass filtered at 3 kHz and stored on a PC using an AD/DA board (ITC-16; HEKA, Lamprecht, Germany) and TIDA software (HEKA). Slices that did not reveal epileptiform activity were discarded from analysis. To test the viability of slices perfused with control ACSF, postsynaptic responses to repetitive electrical stimulation at half stimulation intensity of layers V-VI of the auditory cortex were recorded in layers II-III during the course of the experiment (*Figure 5—figure supplement 1A*; n = 8 slices from eight animals). Note that electrical stimulation and recording were performed in the auditory cortex to exclude manipulations in the S1/M1 region. Analysis of data was performed using MiniAnalysis 4.3.3 (Synaptosoft, Leonia, NJ).

For the replay of S1 and M1 activities, two 3 min recordings from each S1 and M1 in vivo recordings representing typical properties of activity in terms of frequency and distribution of interevent intervals (*Figure 6—figure supplement 1*) were chosen to generate stimulation files. The stimulation files were loaded to the TIDA software (HEKA, Lamprecht, Germany) and delivered to the slices via a stimulation isolation unit and a bipolar tungsten electrode in infragranular layers of the primary motor cortex.

## In vivo electrophysiological recordings

Extracellular in vivo recordings were performed simultaneously from the somatosensory barrel cortex and motor cortex of P6 to P7 wild-type mice.

Experiments performed under urethane anesthesia (1 to 1.5 g/kg bodyweight, Sigma-Aldrich) were done as described previously (*van der Bourg et al., 2016*). After identifying the location of C1 and C2 barrel columns by intrinsic imaging, a silver wire was inserted into cerebellum as a ground electrode. The bone, but not the dura mater, above S1 and M1 were carefully removed. For the electrical recordings, two 4-shank 16-channel electrodes (1–2 MΩ, NeuroNexus Technologies, Ann Arbor, MI) labeled with DiI were inserted perpendicularly into the mapped barrel columns of S1 and into M1 located 1–2 mm posterior to bregma and 0.2–1 mm from midline. The resulting depth of recording sites was 150 to 300 µm from the cortical surface, corresponding to layers II to IV as confirmed by the DiI labeling (*Figure 4A and B*). For S1, this information was substantiated by current source density analysis. Recording sites in S1 were localized above or within the thalamo-receptive layer IV identified as the instant current sink upon sensory stimulation (*Sitdikova et al., 2014*). One hour following electrode insertion, spontaneous activity (multi unit and local field potential signals) was recorded for 1 hr at a sampling rate of 20 kHz using a multi-channel extracellular amplifier and the MC_RACK software (Multi Channel Systems, Reutlingen, Germany).

For recordings under non-anesthetized conditions, 1.5% to 2.5% isoflurane (vaporized in normal air) was initially administered to the pups and 2% lidocain gel (Astrazeneca, Wedel, Germany) was applied at the site of incision. After head fixation, the dose of isoflurane was reduced to 0.4% to 0.8%. Two 4-shank 32-channel electrodes (1–2 MΩ, NeuroNexus Technologies) labeled with DiI were inserted perpendicularly into S1 and into M1 as described above. One hour following electrode insertion, isoflurane was withdrawn. Pups were allowed to recover for at least 20 min from anesthesia before spontaneous activity was recorded for 10 min.

At the end of each experiment, animals were deeply anesthetized with an overdose of ketamine (240 mg/kg) and xylazine (31 mg/kg). Mice were then perfused with sodium phosphate buffer and fixed with 4% PFA. Mouse brains were coronally sectioned at 200 µm thickness and processed for immunhistochemical analysis of electrode position (DiI stain) and acasp3 signal.

## Analysis of in vivo electrophysiological data

Extracellular field potential recordings were analyzed off-line using custom-made routines (https://github.com/agluhmann/Elife-manuscript-2017-/tree/master; copy archived at https://github.com/elifesciences-publications/Elife-manuscript-2017-) in the MATLAB software, version 7.7 (Mathworks, Natick, MA, USA). For each experiment, channels with maximum occurrence of spontaneous events located in S1 and M1 were chosen for further analysis. In each experiment, 180 s continuous recording of spontaneous extracellular local field potential (LFP) without artifact induced by pup movement was chosen for analysis. For the fast Fourier transform (FFT) analysis, LFP recordings were separated into 36 segments. Next, FFT analysis was applied to each 5 s segment with a frequency resolution of 0.2 Hz. Finally, mean FFT was calculated by averaging FFT spectra in all 36 segments. For the event detection, the original trace was downsampled to 1000 Hz and two different sets of bandpass filtering were applied: 4–30 Hz and 30–100 Hz (see *Figure 4—figure supplement 1*). A threshold at 8-fold baseline standard deviation was applied and timestamps of detected troughs in both filtered trace were pooled. Duplication in detection of trough in both filtered data sets were prevented by exclusion of troughs occurring within 8 ms of each other.

## Seizure experiments in vivo

Seizures were evoked by administration of the chemical proconvulsant kainate (2 mg/kg bodyweight, Tocris) intraperitonealy dissolved in 20 µl 0.9% NaCl) (*Puskarjov et al., 2015*). Control mice were injected with the same amount of saline. Animals were constantly monitored after application of the proconvulsant. Myoclonic jerks and tonic-clonic seizures were observed in all kainate-injected mice. Injection at half dose (1 mg/kg bodyweight kainate in NaCl) was repeated upon disappearance of seizures. Mice were sacrificed 5 hr after injection and brains sampled for immunhistochemical analysis.

## Whisker deafferentation

On the day before the experiment, whiskers were clipped on the right side. At the day of experiment, lidocain gel 2% (Astrazeneca) was applied on the right whisker pad every hour for 5 hr. Pups were placed in the litter during each experiment to allow physiological whisker stimulation. After 5 hr, brains were sampled for further immunohistochemical analysis.

## Immunhistochemical analysis

Unspecific binding of antibodies was blocked with 7% normal donkey serum and 0.3% triton diluted in PBS for 2 hr at room temperature (RT) or with 0.1% tri-natrium citrate 0.8% triton diluted in PBS for 1 hr at RT for TUNEL assay. Overnight staining was performed at 4°C with primary antibodies diluted in 2% bovine serum albumin with 0.05% azide and 0.1% triton. For TUNEL, reaction mixture (In Situ Cell Death Detection Kit Fluorescein, Roche Diagnostics GmbH, Mannheim, Germany) was applied for 2 hr at 37°C. Thereafter, cells were washed three times with PBS and incubated with DAPI and Cy3- or DyLight488-coupled secondary antibodies (Dianova and Biomol, Hamburg, Germany) diluted in 2% bovine serum albumin with 0.05% azide in PBS at RT for 2 hr. Specimens were mounted with Fluoromount (Sigma-Aldrich). The following antibodies were used: rabbit polyclonal anti-cleaved Caspase 3 (aCasp3) (Cell Signaling Technology, Inc., San Diego, CA, USA) 1: 200; DAPI (Sigma-Aldrich) 1: 200; guinea pig polyclonal anti-serotoninergic transporter (Synaptic Systems, Goettingen, Germany) 1:200; mouse monoclonal anti-GFAP (Synaptic Systems) 1:500.

## Image analysis

For aCasp3 and TUNEL quantification, images of slices were taken with a 10x objective at an Olympus IX81 epifluorescence microscope and subsequently analysed with ImageJ. Experimenters were blinded for experimental conditions. Based on the DAPI staining, cortical region of interest was divided into six sectors a to f of 0.55 mm width for age group P5-7, with sector a positioned at $0.73 \pm 0.4$ mm from midline. Sector-width was adjusted to the respective brain volume in additional analyzed age groups. For colocalization analysis of aCasp3 and GAD67-GFP signaly or aCasp3 and GFAP signals, pictures were taken with a confocal microscope (TCS SP5, Leica, Wetzlar, Germany). Representative images of single neurons were taken with a confocal microscope (63X, TCS SP5, Leica, Wetzlar, Germany) and processed using Huygens Remote Manager Deconvolution and ImageJ softwares. Maximal projections of minimum of 52 stacks were used to obtain the respective Z-projections.

## Statistical analysis

All data were analyzed with Prism (GraphPad Software) and are expressed as mean ±SEM. p values of less than 0.05 were considered to be statistically significant, whereas p values greater than 0.05 were considered non-significant (n.s.). Experimental groups consisted of at least three biological replicates gathered from minimum 3 independent experiments. Statistical analysis of 2 experimental groups was performed with the parametric two-tailed Student's t-test. If more than 2 groups were compared, an analysis of variance (ANOVA) was performed and differences between groups were analyzed by a subsequent Newman–Keuls post-hoc test. If applicable, two-way ANOVA was performed and replicate means were compared by a subsequent Bonferroni pos-hoc test.

## Acknowledgements

This work was supported by funding from the DFG to HJL and AS (SFB 1080). We thank our colleague Beate Krumm for her excellent technical assistance. Support by the Core Facility Microscopy of the IMB in Mainz is gratefully acknowledged.

## Additional information

### Funding

| Funder | Grant reference number | Author |
|---|---|---|
| Deutsche Forschungsge-meinschaft | Collaborative Research Center 1080 | Anne Sinning Heiko J Luhmann |

The funders had no role in study design, data collection and interpretation, or the decision to submit the work for publication.

### Author contributions

Oriane Blanquie, Conceptualization, Formal analysis, Investigation, Methodology, Writing—original draft, Slice preparations, immunostainings and image analysis, part of field potential recordings; Jenq-Wei Yang, Formal analysis, Investigation, In vivo experiments and analysis; Werner Kilb, Software, Formal analysis, Validation, Investigation, Writing—review and editing, Field potential analysis, stimulation protocol design; Salim Sharopov, Formal analysis, Investigation; Anne Sinning, Conceptualization, Supervision, Funding acquisition, Validation, Methodology, Writing—review and editing; Heiko J Luhmann, Conceptualization, Supervision, Funding acquisition, Validation, Writing—review and editing

### Author ORCIDs

Oriane Blanquie http://orcid.org/0000-0003-2361-7129
Anne Sinning http://orcid.org/0000-0002-1518-7272
Heiko J Luhmann http://orcid.org/0000-0002-7934-8661

### Ethics

Animal experimentation: All experiments were conducted in accordance with National and European (86/609/EEC) laws for the use of animals in research and were approved by the local ethical committee (Landesuntersuchungsamt Rheinland-Pfalz 23.177-07/G 10-1-010).

### Decision letter and Author response

Decision letter https://doi.org/10.7554/eLife.27696.027
Author response https://doi.org/10.7554/eLife.27696.028

## Additional files

### Supplementary files

• Transparent reporting form
DOI: https://doi.org/10.7554/eLife.27696.026

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
