## [Decision Letter]

Thank you for submitting your article "Electrical activity controls area-specific expression of neuronal apoptosis in the developing cerebral cortex" for consideration by *eLife*. Your article has been reviewed by two peer reviewers, and the evaluation has been overseen by a Reviewing Editor and a Senior Editor. The two reviewers, Zoltan Molnar and Rustem Khazipov, have agreed to reveal their identity.

The reviewers have discussed the reviews with one another and the Reviewing Editor has drafted this decision to help you prepare a revised submission.

Summary:

The authors use a combination of immunohistochemistry and electrophysiological techniques from the mouse somatosensory and motor cortices and report that these two areas display a difference in neuroapoptosis during the early neonatal period and provide evidence that this regional difference can be secondary to different activity levels. The experiments and analysis are well performed, the results are interesting and conclusions sound.

Essential revisions:

1) Given the areal differences in aCasp3 staining reported in Figure 2, Figure 1 legend needs to be more precise about the areas imaged and quantified.

2) Figure 1 reports aCasp3+ as cells/mm2 – during this time, the cortex expands considerably, so it would be valuable to have an estimate of cell death/brain volume in order to identify whether the decline in aCasp3+ density observed after P6 is a decrease in total number of cells dying, or a decrease in cell density due to cortical expansion.

3) The authors report that L6 is almost devoid of aCasp3+ cells at P5-P7, and report a relatively higher rate of cell death in L5 of older brains. As they have already stained younger and older material in order to generate Figure 1 and Figure 2, it would be illuminating if the authors could comment on whether L6 has cell death occurring at other ages in the range they tested.

4) Figure 2: to demonstrate the overlap between aCasp3 and TUNEL staining, a higher magnification panel is needed, as the TUNEL image looks virtually unstained (background level signal) in the figure (panel E1).

5) Figure 4: Panel A3 and B3 suggest a similar rate of apoptosis in both pharmacological conditions – yet the images in panels A2 and B2 look like there is much more cell death in sector A/B of B2 than A2 (pharmacological manipulation) – is the B2 image representative of the entire data set?

6) In main text, give indication of number of animals used for whisker clipping/lidocaine experiments.

7) The authors are experts in subplate and layer 6b. Surely they need to give some kind of statement on the apparent lack of apoptotic profiles in layer 6 and especially layer 6b. This layer is largely considered as a transient layer (see Hoerder-Suabedissen and Molnar review). Moreover, it has been demonstrated with combined birthdating and marker expression studies that some layer 6b cells survive in larger number than others (Cereb Cortex. 2013 Jun;23(5):1473-83.). It would be appropriate if the authors could elaborate on these issues in their Introduction and Discussion.

8) The images, especially low-mag views of aCasp3 staining are often not convincing. Higher power images or more representative images may be required.

9) This study focused on the second wave of developmental apoptosis occurring during the brain growth spurt period. However, there is another wave of developmental neuroapoptosis, occurring during embryonic neurogenesis period and this should be indicated in the Introduction.

10) It would be important to estimate the proportion of neurons undergoing apoptosis through the entire population in these two regions.

11) Images presented on Figure 1C1, C2, D1, D2 display some features of the apoptotic nuclear degradation; non-neuronal cells on D2 are also suspicious for nuclear apoptotic bodies. As the authors used Dapi staining, these nuclear changes should be presented.

12) Conclusion that " the critical time window of cell death is comparable over the whole cortex" is an overstatement – as only S1 and M1 were analyzed here.

13) While lower rates of apoptosis in L5 vs. L1-4 are clear in M1 (subsection “Developmental apoptosis affects cortical neurons in an age-, area- and layer-dependent manner”, second paragraph; Figure 2D2), the difference between L5 and L1/4 in S1 is more complex and this should be described in the Results section in more detail (e.g. whether sector d shows a significant difference?).

14) Multishank silicone probe recordings: could the regional differences in the activity levels be due to the use of urethane in these experiments (as urethane may differently affect these two regions)? Also, it is unclear from the results as they are presented from which layer the recordings have been performed, and whether there is any correlation between the apoptosis and the activity in different layers.

15) On a more general note, only few neurons undergo apoptosis despite of activity blockade and only some can be rescued by activity, if one scales the number of apoptotic neurons against the entire population. Also, some neuronal types seem to be well protected from apoptosis and their survival does not seem to depend on activity (such as L6 cells as shown in this report). Is the rule that activity protects against apoptosis universal for all neurons? What are the factors that define particular vulnerability of certain neurons? It would be interesting to know the authors opinion on these questions in the Discussion.

16) The authors should provide some specific statements on subplate cell death in mouse (or the lack of it).

---

## [Author Response]

*1) Given the areal differences in aCasp3 staining reported in Figure 2, Figure 1 legend needs to be more precise about the areas imaged and quantified.*

We included in Figure 1 legend that the quantitative analysis of the density of apoptotic cells was performed “in the mouse primary motor and somatosensory cortices”.

*2) Figure 1 reports aCasp3+ as cells/mm2 – during this time, the cortex expands considerably, so it would be valuable to have an estimate of cell death/brain volume in order to identify whether the decline in aCasp3+ density observed after P6 is a decrease in total number of cells dying, or a decrease in cell density due to cortical expansion.*

We agree with the reviewers that an increase in brain volume could influence the density of apoptotic neurons. In the present study, we have assessed the lateral expansion of the brain by measuring the distance between the midline and the lateral boundary of acute coronal slices (Author response image 1A1). While we found that cortices expand laterally by a factor of 1.2 between P1-3 and P5-7, we did not observe a significant lateral expansion between P5-7 and P9-11 (Author response image 1A2). We also estimated the radial expansion of the cerebral cortex in the 3 age groups (Author response image 1A1 and 1A3). While the thickness of cortical layers I to VI markedly increases between P1-3 and P5-7, we did not find any significant radial growth between P5-7 and P9-11. This is in agreement with previous data showing in mice that the largest expansion of the brain takes place during the 5 first postnatal days (Baloch et al., 2009). Thus, we are confident that the decline in aCasp3-positive neurons density observed between P6 and P10 is not due to brain expansion but reflects a physiological decline in cell death. These results are in agreement with previous data showing a decrease in the density of apoptotic cells per brain volume from P1 to P7 on depending on the brain region (Mosley et al., 2016, Ahern et al., 2013).

To follow the reviewers’ suggestion, we further estimated the number of apoptotic cells/mm³ (Author response image 1). 50 µm slices were imaged using an epifluorescence microscope. All aCasp3-positive neurons can be imaged within one focus plan. Thus, we used 0.05 mm as thickness of our field of view for calculation. However, because cell number/mm² is the unit used throughout the manuscript and because the relative change in apoptotic densities over time is not biased by the unit, we would prefer to keep the graph in aCasp3/mm² in the manuscript.

**Author response image 1. respfig1:** Brain expansion and apoptosis in the developing mouse neocortex. (**A1**) Lateral expansion of developing mouse brains was determined by measuring the distance between midline and lateral boundary of coronal sections (filled arrow). Radial expansion was investigated by measuring the thickness of neocortical layers (open arrow). (**A2**) Mouse neocortex laterally expands from P1-3 to P5-7 by a factor of 1.23 ± 0.06, but does not display a significant lateral expansion between P5-7 and P9-11. (**A3**) Quantification of radial expansion (layers I to VI) shows a significant increase between P1-3 and P5-7 but not between P5-7 and P9-11. (**B**) Quantification of the number of aCasp3-positive cells per mm³ brain volume during the 10 first postnatal days.

*3) The authors report that L6 is almost devoid of aCasp3+ cells at P5-P7, and report a relatively higher rate of cell death in L5 of older brains. As they have already stained younger and older material in order to generate Figure 1 and Figure 2, it would be illuminating if the authors could comment on whether L6 has cell death occurring at other ages in the range they tested.*

We now quantified the number of apoptotic neurons in layer VI of P1-3, P5-7 and P9-11 animals. We found that layer VI displays only a low amount of apoptotic neurons in all 3 age groups and does not exhibit any sector-specificity. These results now appear in Figure 3 (previous Figure 2) and in the text (subsection “Developmental apoptosis affects cortical neurons in an age-, area- and layer-dependent manner”).

*4) Figure 2: to demonstrate the overlap between aCasp3 and TUNEL staining, a higher magnification panel is needed, as the TUNEL image looks virtually unstained (background level signal) in the figure (panel E1).*

We now changed the magnification of the TUNEL and the respective aCasp3 stainings (Figure 3, previous Figure 2). Please note that TUNEL signal is less prominent than aCasp3 signal as it only labels the nucleus of cells within a shorter time window (Thomaidou et al., 1997, Collins et al., 1997, Golbs et al., 2007).

*5) Figure 4: Panel A3 and B3 suggest a similar rate of apoptosis in both pharmacological conditions – yet the images in panels A2 and B2 look like there is much more cell death in sector A/B of B2 than A2 (pharmacological manipulation) – is the B2 image representative of the entire data set?*

We now replaced low magnification images for higher magnification pictures and chose examples which are representative of the data set (Figure 5, previous Figure 4).

*6) In main text, give indication of number of animals used for whisker clipping/lidocaine experiments.*

A total number of 10 mice were used for whisker clipping/lidocaine experiments (subsection “Altered electrical activity affects rates of apoptosis and reduces regional differences in vivo”, second paragraph).

*7) The authors are experts in subplate and layer 6b. Surely they need to give some kind of statement on the apparent lack of apoptotic profiles in layer 6 and especially layer 6b. This layer is largely considered as a transient layer (see Hoerder-Suabedissen and Molnar review). Moreover, it has been demonstrated with combined birthdating and marker expression studies that some layer 6b cells survive in larger number than others (Cereb Cortex. 2013 Jun;23 (6)[5]:1473-83.). It would be appropriate if the authors could elaborate on these issues in their Introduction and Discussion.*

In addition to the quantitative analysis of neuronal apoptosis performed in layer VI in the 3 age groups (Figure 3, previous Figure 2), we now discuss the fact that (1) [1]aCasp3 labeling marks apoptotic neurons during a relatively short time window, (2) [2]the subplate layer represents only a thin layer within layer VI and (3) [3]several reports have demonstrated that a substantial portion of subplate neurons may survive beyond this early postnatal period (Discussion, sixth paragraph).

*8) The images, especially low-mag views of aCasp3 staining are often not convincing. Higher power images or more representative images may be required.*

We have now changed the relevant figures (Figure 3 and Figure 5). We have added close-up to the low magnification pictures in Figure 3 (previous Figure 2) and replaced low magnification pictures for high magnification pictures in Figure 5 (previous Figure 4).

*9) This study focused on the second wave of developmental apoptosis occurring during the brain growth spurt period. However, there is another wave of developmental neuroapoptosis, occurring during embryonic neurogenesis period and this should be indicated in the Introduction.*

We extended the Introduction and included that a first peak of apoptosis occurs during embryonic neurogenesis (Introduction, first paragraph).

*10) It would be important to estimate the proportion of neurons undergoing apoptosis through the entire population in these two regions.*

We agree with the reviewer that estimating the proportion of neurons undergoing apoptosis through the entire population would be highly interesting. However, as depicted in Author response image 2, it is very challenging to quantify either the number of nuclei (DAPI) or the number of neurons (NeuN) using the magnification used throughout the manuscript (Author response image 2A1 and 2B1) as well as at a higher magnification (Author response image 2A2 and 2B2). The difficulty is not due to the quality of the immunostaining but rather to the high cell density of layers II-IV. In addition, because aCasp3 staining labels cells only during two to three hours, a co-immunostaining with a neuronal marker would underestimate the actual portion of neurons undergoing apoptosis. Other studies have quantified the total number of neurons and cells within the rodent brain using methods such as isotropic fractionator (Bandeira et al., 2009, Herculano-Houzel et al., 2006, Collins et al., 2010) or design-based stereology (Schmitz and Hof, 2005). However, such studies are beyond the scope of the present study.

**Author response image 2. respfig2:** Immunostaining against DAPI and NeuN in coronal cortical slices from P6 mouse brains. (**A1**) In the S1 region, immunostaining against DAPI and NeuN respectively allows the detection of nuclei in the overall cortical population and the cytoplasm of neurons. The high cell density, in particular in layers I-IV, hampers any reliable quantification. (**A2**) Also at a higher magnification, quantification of NeuN-positive cells is difficult. (**B**) Same than (**A**) but in M1.

*11) Images presented on Figure 1C1, C2, D1, D2 display some features of the apoptotic nuclear degradation; non-neuronal cells on D2 are also suspicious for nuclear apoptotic bodies. As the authors used Dapi staining, these nuclear changes should be presented.*

Because we chose only mildly degraded neurons for representative images and analysis of cell-type specificity in Figure 1, these apoptotic neurons do not display signs of striking nuclear degradation in DAPI signal (Author response image 3). However, to follow the suggestion of the reviewer, we have now included a close-up of DAPI staining of typical aCasp3-positive cells in Figure 2. In these representative images, an aCasp3-positive neuron with an intact nucleus (Figure 2C1) as well as aCasp3-positive neuron displaying a typical nuclear degradation (Figure 2C2) are shown in the two regions of interest. We further stress in the text (subsection “Developmental apoptosis affects cortical neurons in an age-, area- and layer-dependent manner”, second paragraph) and Figure 2 legend that aCasp3-labeling allows the detection of apoptotic neurons at various stages of nuclear and morphological degradation.

**Author response image 3. respfig3:** Characterization of neuronal apoptosis in the developing mouse neocortex in vivo. (**A1**) Most of aCasp3-positive neurons (green) are GAD67-negative, putative glutamatergic neurons. (**A2**) Acasp3 allows the detection of dying neurons at an early stage of apoptosis, when acasp3 signal is not yet translocated into the nucleus and when the nucleus does not display any visible sign of degradation. (**B1**) A subset of apoptotic neurons is expressing the GABAergic marker GAD67 (red). (**B2**) The aCasp3-positive neuron displays aCasp3 signal into the nucleus (green), the nucleus starts to slightly condense (DAPI, blue) and starts to show signs of neurite degradation.

*12) Conclusion that " the critical time window of cell death is comparable over the whole cortex" is an overstatement – as only S1 and M1 were analyzed here.*

We modified the text for “the critical time window of cell death takes place between P5 and P7 in all layers of M1 and S1”.

*13) While lower rates of apoptosis in L5 vs. L1-4 are clear in M1 (subsection “Developmental apoptosis affects cortical neurons in an age-, area- and layer-dependent manner”, second paragraph; Figure 2D2), the difference between L5 and L1/4 in S1 is more complex and this should be described in the Results section in more detail (e.g. whether sector d shows a significant difference?).*

In addition to the previous statement that the difference in density of apoptotic neurons is layer-dependent (two-way ANOVA, layer p=0.0002), we now explicitly state that the density of apoptotic neurons is significantly higher in layers I-IV compared with the remaining layers in sector a (post-hoc test sector a <0.001) but not in the remaining sectors (post-hoc test remaining sectors p>0.05) (subsection “Developmental apoptosis affects cortical neurons in an age-, area- and layer-dependent manner”). We further mention a clear tendency of higher cell death rates in layer V compared with the other layers in sectors c/d (post-hoc test >0.05).

*14) Multishank silicone probe recordings: could the regional differences in the activity levels be due to the use of urethane in these experiments (as urethane may differently affect these two regions)? Also, it is unclear from the results as they are presented from which layer the recordings have been performed, and whether there is any correlation between the apoptosis and the activity in different layers.*

To answer the important question of the reviewer, we performed additional experiments to compare S1 and M1 activity in non-anesthetized animals. These experiments revealed that in non-anesthetized animals, the frequency of detected troughs in S1 (Figure 4E3; 159.73 ± 23.6 min^-1^) and in M1 (51.9 ± 23.4 min^-1^, n=5 animals) is comparable to what is observed in urethane-anesthetized animals (Figure 4D3; 156 ± 27.8 and 63 ± 20 min^-1^, respectively) and more importantly the trough rate is also higher in S1 compared to M1 (Mann Whitney test p=0.03). For the spectral power, we also observed significantly higher values in S1 compared to M1 in non-anesthetized mice (Figures 4E1 and 4E2; two-way ANOVA, cortical area p=0.0015). Furthermore, the spectral power in non-anesthetized mice is significantly higher than in urethane-anesthetized mice for S1 recordings (two-way ANOVA, anesthesia p=0.04) but not different for M1 recordings (two-way ANOVA, anesthesia p>0.05). Overall, these results indicate that also in non-anesthetized animal, activity levels in S1 are significantly higher than that in M1. These results are now added to the manuscript (subsection “Differences in electrical activity of M1 and S1 in vivo parallels area-specific apoptosis”, first paragraph) and displayed in Figure 4 (previous Figure 3).

All recordings from S1 and M1 were recorded in layers II-III to layer IV, as identified from the penetration depth of the electrodes. For S1, this information was substantiated by current source density analysis. Recording sites in S1 were localized above or within the thalamoreceptive layer IV identified as the instant current sink upon sensory stimulation. We included this information in the revised version of the manuscript (subsection “Differences in electrical activity of M1 and S1 in vivo parallels area-specific apoptosis”, first paragraph and subsection “In vivo electrophysiological recordings”, second paragraph).

We next addressed the question, whether rates of apoptosis and activity levels correlate across the different layers. In M1 (sectors a-b; Figure 3A2, previous Figure 2D2), the density of apoptosis is significantly higher in layers I-IV than in the deeper layers while in S1 (sectors c-f), the layer-specificity is more complex (see point 13). In order to test the correlation between rates of apoptosis and activity levels across neocortical layers, we therefore compared the spontaneous in vivo activity in non-anesthetized with the cell death density in M1 only. We found that the summation of spectral power is not significantly lower in layers I-IV compared to deeper layers, but rather tends to be higher (Author response image 4; one-way ANOVA, layers p=0.10). Thus, while levels of activity and rates of apoptosis in layers I-IV are negatively correlated between M1 and S1, this is not the case across cortical layers of M1. We discussed this interesting result in the manuscript (Discussion, fifth paragraph).

**Author response image 4. respfig4:** Density of apoptotic neurons and levels of activity in M1 positively correlate across cortical layers. (**A**) In M1, the density of apoptotic neurons is significantly higher in layers I-IV than in the deeper layers. (**B**) The spectral power tends to be higher in layers I-IV than in layers V and VI.

*15) On a more general note, only few neurons undergo apoptosis despite of activity blockade and only some can be rescued by activity, if one scales the number of apoptotic neurons against the entire population. Also, some neuronal types seem to be well protected from apoptosis and their survival does not seem to depend on activity (such as L6 cells as shown in this report). Is the rule that activity protects against apoptosis universal for all neurons? What are the factors that define particular vulnerability of certain neurons? It would be interesting to know the authors opinion on these questions in the Discussion.*

While the effect of elevated neuronal activity dramatically reduces the number of neurons undergoing apoptosis already after 4 hours, the effect of chronic activity blockade by TTX seems more moderate. Cell-type specific differences in vulnerability to activity modulations have been shown previously, e.g. in Cajal Retzius neurons (Blanquie et al. 2016), and likely also occur in other neuronal subpopulations. In the present study, we focused on functional region-dependent differences in the total neuronal population of layer I-IV. Whether neuronal activity differentially affects neurons in different neocortical layers, i.e. regions of different cellular compositions, remains to be investigated (see point 14). This interesting point is now raised in the Discussion (fourth and fifth paragraphs).

*16) The authors should provide some specific statements on subplate cell death in mouse (or the lack of it).*

We agree that the low rate of apoptosis in the layer VI is interesting (see point 7). This issue is now addressed is the Discussion (sixth paragraph).